# Effects of Methyl Terminal and Carbon Bridging Groups Ratio on Critical Properties of Porous Organosilicate Glass Films

**DOI:** 10.3390/ma13204484

**Published:** 2020-10-10

**Authors:** Alexey S. Vishnevskiy, Sergej Naumov, Dmitry S. Seregin, Yu-Hsuan Wu, Wei-Tsung Chuang, Md Rasadujjaman, Jing Zhang, Jihperng Leu, Konstantin A. Vorotilov, Mikhail R. Baklanov

**Affiliations:** 1Research and Education Center “Technological Center”, MIREA—Russian Technological University (RTU MIREA), 119454 Moscow, Russia; d_seregin@mirea.ru (D.S.S.); vorotilov@mirea.ru (K.A.V.); m_baklanov@hotmail.com (M.R.B.); 2The Leibniz Institute of Surface Engineering (IOM), 04318 Leipzig, Germany; sergej.naumov@iom-leipzig.de; 3Department of Materials Science and Engineering, National Chiao Tung University, Hsinchu 30049, Taiwan; Shiuan5276@yahoo.com.tw (Y.-H.W.); jimleu@nctu.edu.tw (J.L.); 4National Synchrotron Radiation Research Center, Hsinchu 30076, Taiwan; weitsung@nsrrc.org.tw; 5Department of Microelectronics, North China University of Technology, Beijing 100144, China; rasadphy@duet.ac.bd (M.R.); zhangj@ncut.edu.cn (J.Z.); 6Department of Physics, Dhaka University of Engineering & Technology, Gazipur 1707, Bangladesh

**Keywords:** organosilicate glass, low-*k* films, carbon bridges, pore structure, ellipsometric porosimetry, FTIR, density functional theory (DFT), Young’s modulus, pore morphology, GISAXS

## Abstract

Organosilicate glass-based porous low dielectic constant films with different ratios of terminal methyl to bridging organic (methylene, ethylene and 1,4-phenylene) groups are spin-on deposited by using a mixture of alkylenesiloxane with organic bridges and methyltrimethoxysilane, followed by soft baking at 120–200 °C and curing at 430 °C. The films’ porosity was controlled by using sacrificial template Brij^®^ L4. Changes of the films’ refractive indices, mechanical properties, *k*-values, porosity and pore structure versus chemical composition of the film’s matrix are evaluated and compared with methyl-terminated low-*k* materials. The chemical resistance of the films to annealing in oxygen-containing atmosphere is evaluated by using density functional theory (DFT). It is found that the introduction of bridging groups changes their porosity and pore structure, increases Young’s modulus, but the improvement of mechanical properties happens simultaneously with the increase in the refractive index and *k*-value. The 1,4-phenylene bridging groups have the strongest impact on the films’ properties. Mechanisms of oxidative degradation of carbon bridges are studied and it is shown that 1,4-phenylene-bridged films have the highest stability. Methylene- and ethylene-bridged films are less stable but methylene-bridged films show slightly higher stability than ethylene-bridged films.

## 1. Introduction

Low dielectric constant (low-*k*) materials and low resistivity metals (Cu, Co, Ru etc.) have been introduced into microelectronics technology to replace traditional SiO_2_ and Al since the late 1990s. The implementation of Cu and low-*k* dielectrics was required to reduce the signal propagation delay (resistance-capacitance *RC*-delay) in interconnection structures of ultra large-scale integration (ULSI) devices, and to reduce power dissipation and cross-talk noise between the metal lines [1,2]. After extensive evaluation of different low-*k* candidates from organic polymers to metal–organic frameworks [3], organosilicate glass (OSG)-based films have been recognized as the most suitable low-*k* materials for the present ULSI technology. In these materials, bridging oxygen atoms of silica-like matrix are partly replaced by terminal organic (methyl) groups. The organic terminal groups make porous OSG materials hydrophobic and make it possible to minimize or avoid adsorption of water molecules that have high dielectric constants (the *k*-value of water is about 80). The adsorbed water molecules also degrade the device reliability, increasing the leakage current and reducing the breakdown field. To reduce the dielectic constant, these materials have to be made porous with the terminal methyl groups mainly located on the pore wall surfaces [4].

Recently, OSG low-*k* materials with crosslinking (bridging) carbon groups have become more popular. The materials with bridging carbon (organic) groups can form films with ordered porosity and they are termed as periodic mesoporous organosilicates (PMO) [5]. The incorporated bridging alkyl groups should improve mechanical properties because of the higher bending rigidity of Si–C–Si bonds than that of Si–O–Si [6,7]. The improved strength of these films is important for the back-end-of-line (BEOL) integration technology and reliability of ULSI devices, especially during the final device packaging [8]. Different types of PMO materials have already been explored as low-*k* candidates. More detailed information related to these efforts can be found in the review paper [5] and references used there.

However, the terminal and bridging carbon groups in OSG films have different affinities to adsorption of water molecules. The bridging carbon groups are not able to make OSG materials sufficiently hydrophobic like terminal methyl groups. Therefore, the simultaneous incorporation of terminal groups and bridging carbon groups is needed. Successful implementation of these types of low-*k* materials needs careful selection of the most suitable bridging groups and the optimal relation between concentrations of the terminal and bridging carbon groups. However, the results of systematic comparative study of PMO materials with different ratios of terminal and bridging carbon groups are not known. Taking this into account, the goal of this research has been the development and deposition of low-*k* films with different types of bridging groups (methylene, ethylene, 1,4-phenylene) and different concentrations of terminal methyl groups (Figure 1), and comparative analysis of their properties. The concentration of terminal methyl groups precursor (methyltrimethoxysilane) varied from 100% to zero. The first part of this work is experimental: the deposited films were evaluated by using FTIR spectroscopy (chemical composition), ellipsometric porosimetry (EP: refractive index, porosity and pore radius distribution), grazing-incidence small-angle X-ray scattering (GISAXS: pore structure), and nanoindentation (mechanical properties). The second part of the work reports results of ab initio density functional theory (DFT) calculations to understand the nature of chemical reactions during the interaction of carbon bridges with residual oxygen (for instance, during thermal curing or integration processes). These results are also important for the selection of the materials most suitable for integration into ULSI devices.

## 2. Experimental Design, Materials and Methods

The OSG samples with different types of carbon bridges (methylene, ethylene and 1,4-phenylene) for comparative analysis were prepared by using commercially-available precursors and special procedures designed for this purpose. The film-forming solution used acid-catalysed (HCl, 37%, Sigma-Aldrich, St. Louis, MO, USA) condensation of methyl-modified silicon alkoxide (methyltrimethoxysilane, MTMS, >98%, Fluka, Buchs, Switzerland) and alkylenesiloxane in tetrahydrofuran (THF, anhydrous, ≥99.9%, Sigma-Aldrich, St. Louis, MO, USA) in the presence of deionised water. The alkylenesiloxane contains different organic bridges between the silicon atoms: methylene (1,2-bis(triethoxysilyl)methane, BTESM, 97%, abcr, Karlsruhe, Germany), ethylene (1,2-bis(trimethoxysilyl)ethane, BTMSE, 96%, Sigma-Aldrich) or 1,4-phenylene (1,4-bis(triethoxysilyl)benzene, BTESB, 95%, abcr). Alkylenesiloxanes (BTESM, BTMSE or BTESB) were mixed with MTMS in molar concentrations equal to 0, 25, 45, 60 and 100 mol% (Figure 1). Appropriate amounts of MTMS, alkylenesiloxane, THF, water, and the acid were added to a flask fitted with a magnetic stirrer. The water content per methoxy group in the solution was 0.4. The MTMS:HCl mole ratio was 1:0.001. The final equivalent Si content in the solutions was 5.3–6.7 wt%. Porogen Brij^®^ L4 (C_12_H_25_(OCH_2_CH_2_)_4_OH) from Sigma-Aldrich, with a molar mass of 362 g/mol, was used as a template in the evaporation-induced self-assembly (EISA) process to control the films’ porosity during the spin-on deposition [9]. The porogen concentration was kept equal to 30 wt% to the sum of metal alkoxide and alkylenesiloxane. The films deposited on the top of Si wafers were soft baked on a hot plate at *T_a_* = 120 °C, 30 min, then *T_a_* = 200 °C, 30 min, and the final curing (hard bake) was carried out at *T_a_* = 430 °C, 60 min. The uniformity and homogeneity of the films are confirmed by the UV-Vis (SENTECH, Berlin, Germany) absorption spectra and SEM (Field Electron and Ion Company (FEI), Hillsboro, OR, United States) are given in Appendix A, respectively.

Chemical composition of the deposited films was evaluated by using Nicolet 6700 Fourier-transform infrared (FTIR) spectrometer (Thermo Fisher Scientific, Waltham, MA, USA) in transmission mode with a resolution of 4 cm^−1^ (at least 64 scans) in the extended range of 7400–400 cm^−1^ with nitrogen purging. The baseline was fitted using the ~10th-order polynomial function combined with the sigmoidal function for the inflection region, ranging from 1200 to 1100 cm^−1^. Background spectra were obtained using a pure silicon sample cut from the same wafer that was used for the film deposition. All spectra are normalized to the highest Si–O–Si peak (imposition of network and networked suboxide bands).

The thickness and index of refraction (RI), porosity, and pore size distribution of the films were estimated by using ellipsometric porosimetry (EP) [10] based on an in situ spectroscopic ellipsometer SE-850 (SENTECH, Berlin, Germany) with λ = 300–800 nm. Isopropyl alcohol (IPA) vapor with N_2_ carrier gas was used as an adsorptive. The films’ open porosity is calculated as the volume of condensed liquid IPA from RI values measured during the adsorption (*n_eff_*) by using a modified Lorentz–Lorenz equation [10]:(1)$$V=\frac{\left(\frac{n_{eff}^{2}-1}{n_{eff}^{2}+2}\;-\frac{n_{s}^{2}-1}{n_{s}^{2}+2}\right)}{\left(\frac{n_{ads}^{2}-1}{n_{ads}^{2}+2}\;-\frac{n_{s}^{2}-1}{n_{s}^{2}+2}\right)},$$
where *n_eff_* is the measured RI of porous film filled by adsorptive molecules, *n_ads_* is the RI of liquid adsorptive (IPA in our case), *n_s_* is the RI of the film skeleton (the dense part of the film). The calculation of the pore size of mesoporous film is based on the Kelvin equation [11]. The size of the micropores is calculated by using the Dubinin–Radushkevich equation adapted for EP [11].

The film shrinkage Δ*d* was evaluated as the change in film thickness (%) after the soft bake at 200 °C and after the hard bake at 430 °C: $∆d=\frac{d_{200}-d_{430}}{d_{200}}\times 100\%$.

The pore geometries and ordering were characterized by GISAXS (National Synchrotron Radiation Research Center, Hsinchu, Taiwan) using beamline BL23A (National Synchrotron Radiation Research Center, Hsinchu, Taiwan) in the National Synchrotron Radiation Research Center in Taiwan. The incidence beam was monochromated to a wavelength (λ) of 1.55 Å with Δλ/λ ≈ 10^−3^ resolution. The detailed information of the detector, the scattering wave vector (*q*) range, incident angle and X-ray energy had been described in [12]. All GISAXS data were corrected for sample transmission, background, and the detector sensitivity. Then the pore geometries were analyzed and simulated using a cylinder model [13] based on SASView software (Caltech, Pasadena, CA, USA) [14].

Nanoindentation (NI) was used to measure the Young’s modulus (YM) of the films. The measurements were conducted using a nanoindenter (MTS Nano Indenter XP System, MTS Systems Corporation, Eden Prairie, MN, USA) in a continuous mode. A pyramidal Berkovitz tip was used to obtain the reduced modulus (*E_r_*). The Oliver–Pharr method was used to determine the elastic modulus (*E*) [15]. In addition, the indentation depth was kept smaller than 20% of the film thickness to eliminate the influence of the Si substrate.

To measure the dielectric constant, capacitance-voltage (CV) and capacitance-frequency (Cf) measurements were carried out using the MDC CSM/Win Semiconductor Measurement System (Materials Development Corporation, Chatsworth, CA, USA). The test system involves the MDC 802B-150 (Materials Development Corporation, Chatsworth, CA, USA) mercury probe station with the contact diameter of about 790 μm and an Agilent 4284A inductance, capacitance and resistance meter (Santa Clara, CA, USA). For these measurements, the silicon wafers have a resistivity of 0.005 Ohm-cm. The dielectric constant was estimated in the frequency range of 1 to 100 kHz.

Additional information related to resistance of the deposited films to thermal annealing, and the mechanisms of their possible destruction by residual oxygen, was obtained by density functional theory (DFT). This information is important for optimization of the curing processes and also sheds a light on the compatibility with integration processes. The DFT calculations were based on application of the PBE0-D3 functional [16,17,18], which includes London dispersion interactions [19] at the PBE0-D3/6-31G** level of theory, which is implemented in the Jaguar 9.6 program [20]. The total enthalpy (*H*) and Gibbs free energy (*G*) were obtained from frequency calculations. The difference of the calculated *H* and *G* between the reactants and products are presented as the reaction enthalpies (Δ*H*) and Gibbs free energies (Δ*G*).

## 3. Results and Discussion

### 3.1. Chemical Composition of Methylene, Ethylene and 1,4-phenylene-bridged OSG Films

Figure 2 and Figure 3 show FTIR spectra of completely cured porous OSG films. Materials with 45 mol% of bridge concentration (Samples 45M, 45E and 45B in the Table 1) are selected for demonstration as typical representatives. The most intensive peaks in FTIR spectra are related to the silica-like matrix (Si–O–Si stretching vibrations at 1200–1000 cm^−1^) and to the terminal Si–CH_3_ groups (~1275 cm^−1^) typical for various OSG low-*k* films [21,22,23]. The peaks associated with the bridging groups are not so pronounced (1700–1300 cm^−1^) due to their small concentration and much smaller absorption coefficient in comparison with Si–O bonds. It is clearly seen that the spectrum of the Sample 45B has the largest number of distinctive absorption bands due to the presence of 1,4-phenylene (p-disubstituted) rings in their structure as bridges. The following characteristic bands can be noted: 3050, 1600, 1510, 1150, 1025, 525 cm^−1^ [24,25].

The FTIR spectra of the 45M and 45E samples have much less difference. The characteristic band of Sample 45E is located at ~720 cm^−1^, which is correlated to the C–C bonds of ethylene bridges [26,27,28]. The distinctive peak of C–H bonds of methylene bridges in Sample 45M is located at ~1360 cm^−1^ [29], whose position differs markedly from the peak of C–H bonds of ethylene bridges in Sample 45E (~1415 cm^−1^) [30]. Some differences are also visible in the 3100–2800 cm^−1^ region. The most important one is the high intensity of CH_2_ peaks at 2950–2850 cm^−1^ in ethylene-bridged films. The largest intensity of Si–OH at ~950 cm^−1^ [31] is observed in Sample 45B, and the smallest is in 45M (the complete set of data is summarized in the Appendix A, Appendix A). In addition, the Sample 45B contains more surface silanol groups and adsorbed water. The surface silanols are the centers of water molecules’ adsorption (wide band at 3600–3200 cm^−1^). A similar conclusion can be made from the dielectric constant and water contact angle data, which are presented in Appendix A, Appendix A and Appendix A, respectively. The spectra of other series with 25, 60 and 100 mol% of bridge concentration (Table 1) show that the increase in the alkylenesiloxane precursors’ concentration increases the intensity of bridging components and reduces SiCH_3_ bonds’ concentration, but they are basically similar to the spectra presented in the Figure 2 and Figure 3 (the full set of spectra is available on reasonable request to the authors).

### 3.2. Impact of Bridge Type and Composition on the Properties of Carbon-bridged Low-k Films

Table 1 summarizes comparative data for OSG samples with different types and concentrations of carbon bridges. The sample in Experiment 1 was obtained by using only methyltrimethoxysilane (MTMS). The resulting film is a porous methyl-terminated OSG known as (poly)methylsilsesquioxane or MSSQ [26,32], which has a regular ladder-like structure [33]. The Samples 25M, 45M, 60M (Experiments 2–4) were deposited from mixture of MTMS and BTESM with different ratios and contain both terminal methyl and bridging methylene groups. Similarly, the Samples 25E, 45E, 60E were deposited from mixture of MTMS/BTMSE, while the Samples 25B, 45B, 60B (Experiments 2–4) were produced by using MTMS/BTESB, and contain ethylene and 1,4-phenylene bridges, respectively. The Samples 100M, 100E, 100B (Experiment 5) were deposited using pure alkylenesiloxanes without MTMS. As already mentioned, the porogen concentration was kept as 30 wt% of the sum of alkyl- and alkylenesiloxane in all cases. The samples from the Experiment 5 containing only bridging carbon groups were found to be hydrophilic. Hydrophilicity of these films and fast moisture adsorption made the reliable measurement of the dielectric constant by mercury probe impossible. Similar problems were also observed for samples in the Experiment 4 (Samples 60M, 60E, 60B), suggesting that the concentration of hydrophobic terminal methyl groups must exceed a certain critical level if the films are targeted for microelectronics application. 

The data presented in Table 1 provide important information about properties of the films versus type and concentration of the bridging groups. Figure 4 shows the change of refractive indices of the films and their skeleton versus the type and concentration of the bridging groups. RI values increase with the concentration of carbon-based bridging groups and they are all larger than the RI of methyl-terminated film. In the case of methylene and ethylene bridges, these changes are not significant, but a much stronger change is observed in the case of the 1,4-phenylene bridge. This effect is obviously related to the reduced porosity and higher molecular polarizability of aromatic rings in comparison with aliphatic compounds. The change of skeleton RI calculated from EP data shows the same tendency. Therefore, it is reasonable to conclude that the skeleton properties play a more important role; thereby, the impact of molecular polarizability on RI values is generally dominant. 

Similar behavior demonstrates the *k*-value of the films measured in Experiments 1–3 (Figure 5). The *k*-value increases with the concentration of the bridging groups. The possible reasons are distortion of the regular ladder-like structure of MSSQ films with the reduction in porosity, increase in residual silanol groups’ concentration and change of volume polarizability (phenylene bridged films) [35]. This effect is most pronounced in the films with the 1,4-phenylene bridge. Unfortunately, as already mentioned, reliable measurements of the *k*-values of the samples deposited in the Experiments 4 and 5 were not possible because of their hydrophilic nature and the presence of adsorbed water.

Figure 6 shows the Young’s modulus values measured by nanoindentor. One can see that YM increases with the concentration of the bridging groups. Moreover, the YM values for the 1,4-phenylene-bridged films are higher than those of methylene and ethylene-bridged films, except that of the sample prepared from 60 mol% of BTMSE (60E). The Young’s moduli of the films deposited from alkylenesiloxane/MTMS mixtures show almost linear dependence on the bridge concentration in the range of alkylenesiloxane concentration from zero to 60 mol%. For additional comparison, Figure 6 also shows three data points reported for well-known low-*k* films (LKD-5109, MSQ2.4 and NCS2.3) with terminal methyl groups. It is remarkable that the structure of methyl-terminated films can be significantly optimized (even YM = 10 GPa for modified NCS was reported by Fujitsu [36]) and Young’s modulus reaches the values comparable with ~25 mol% carbon bridge concentration.

### 3.3. Porosity and Pore Size Distribution (EP Data)

Figure 7 shows the adsorption and desorption isotherms of IPA vapors and calculated pore radius distribution in low-*k* films prepared from only (a) MTMS, (b) BTESM, (c) BTMSE and (d) BTESB at a fixed 30 wt% porogen content. One can see that the porosity of the films reduces from 39% in the sample (a) to 20% in the sample (d). The sample (d) containing 1,4-phenylene bridges demonstrates the smallest pore size. This film is completely microporous with the pore radius of 0.76 nm. The samples (b) and (c) containing only methylene or ethylene bridges, respectively, demonstrate bimodal porosity. They contain micropores similar to those observed in the sample (d) but also contain larger mesopores. The sample (a), containing only terminal methyl groups, has completely different adsorption and desorption isotherms. It can be classified as a representative of the type H2 isotherm that is typical for the pores with narrow interconnecting mouths (ink-bottle pores) and internal voids [39]. The size of interconnecting mouths is represented by a desorption curve while the adsorption branch represents the size of internal voids [40]. The graphs clearly demonstrate the effects of bridging groups on the pore size and porosity. The 1,4-phenylene bridge has the strongest effect, while the difference between the films with methylene and ethylene bridges is not significant.

The samples with different concentrations of methyl terminals and different bridges have been studied in detail. The completed set of data can be found in the Appendix A, Appendix A, and here we only demonstrate the effect of 1,4-phenylene bridges because of the most pronounced impact of composition (Figure 8). The sample with the highest concentration of 1,4-phenylene bridges (MTMS/BTESB ratio equal to 40/60) shows isotherms quite similar to the film prepared from pure BTESB (Figure 7d). It has exactly the same pore size distribution (0.76 nm at maximum probability) but the porosity is slightly higher (27% instead of 20% in pure BTESB). The increase in MTMS content (curves (b) and (c) in Figure 8) increases porosity, and the pore radius distribution becomes bi-modal with the presence of mesopores. However, the isotherms are still quite different than the MSSQ film ones (Figure 7a).

Figure 9 shows the summary of all EP data. One can see that generally the carbon bridge concentration reduces both porosity and pore size. The difference between the films with methylene and ethylene bridges is not so pronounced while the 1,4-phenylene-bridged samples show a clear difference. The Samples 45B, 60B and 100B have the smallest pore sizes. The porosity of these films reduces with the concentration of 1,4-phenylene bridges and partly leads to the increase in *k*-value, as shown in the Figure 5. Therefore, Figure 9 suggests that Sample 45B is more interesting for certain technological applications when the pore size can be kept very small at relatively high porosity.

### 3.4. Pore Geometries Characterized by GISAXS

Two dimensional (2D) GISAXS scattering patterns for OSG films hard baked for 30 min at 400 °C in nitrogen are shown in Figure 10a–m. The MSSQ films (Figure 10a) show isotropic, well defined rings of maxima scattering pattern, which indicate the existence of randomly distributed pores with narrow pore size distribution. In addition, no pore correlation was observed, as illustrated in the scattering pattern [41]. Most of films shown in the Figure 10b–g show similar isotropic and well defined rings of maxima scattering pattern with a few exceptions such as 25M [Figure 10b], 25E [Figure 10c], 25B [Figure 10d], and 45E [Figure 10f], which exhibit a well-defined ring of maxima scattering pattern with a strong horizontal correlation, at a large angle, i.e., at a higher *q*. This correlation peak indicates the existence of ordered pores in the *x*-direction, or film planes with a specific pore correlation, or pore-to-pore distance. 

Pore geometries can be analyzed and obtained from the scattering patterns by using the GISAXS model. The scattering pattern intensity, *I*_(*q*)_, is proportional to the product of the intra-particle structure factor (form factor), *P*_(*q*)_, and inter-particle structure factor (structure factor), *S*_(*q*)_, as expressed by Equation (2) [13]:(2)$$I_{\left(q\right)}\propto P_{\left(q\right)}\cdot S_{\left(q\right)},$$
where *q* is the scattering wave vector defined by the wavelength of X-ray radiation (λ) and scattering angle (*θ*), as expressed by Equation (3):(3)$$q=\frac{4\pi }{\lambda }\cdot \sin\theta$$

The structure factor *S*_(*q*)_ is close to 1 in a system without inter-particle interactions or a low-concentration system, and thus can be ignored. The average distance between pores (*D*) (pore wall thickness) is determined from the intensity maximum (*q_xy_*_, *max*_), which belongs to the structure factor *S*_(*q*)_. Thus,
(4)$$D=\frac{2\pi }{q_{xy}}$$

To examine the pore correlation (pore-to-pore distance) the GISAXS in-plane intensity versus *q_xy_* (Å^−1^) are plotted for the hybrid OSG films (25-, 45-, and 60-M, -E, and -B) in Figure 11a–c, respectively. Their average pore-to-pore distance, *L*, is summarized in Table 2. When the loading of bridge components is 25 mol%, short-range in-plane pore ordering is observed for all three bridged films (25M, 25E, and 25B) with a local maximum. Furthermore, *L*_25M_ (19.0 nm) > *L*_25E_ (10.8 nm) > *L*_25B_ (8.6 nm). When the loading of bridge components increases to ≥45 mol%, samples become more random due to a loss of clear correlation peak, except 45E still retaining an in-plane short-range pore ordering, with a *L*_45E_ of 8.1 nm.

The pore shape and size are then simulated and analyzed using SASView software. Modelling is based on the distribution of the scattering intensity, which is proportional to the ensemble average of the form factor, *P*_(*q*)_, in a polydisperse system [21]. A variety of particle shapes (spheres, elliptic, or cylinder etc.) were simulated for 45M and illustrated in Figure 12. The scattering intensity profile of 45M is best fitted and described by a cylindrical shape rather than an elliptic or spherical shape. Subsequently, the pore diameter and length for all OSG films are further calculated from the fitting curve versus *q_xy_* based on cylindrical shape. To be precise, the shape of the pores is a flat disc because the diameter of the cylinder is larger than its length, i.e., the aspect ratio of diameter to length is more than 1. The pore diameters, pore lengths, and diameter/length ratio of these OSG films are summarized in Table 2. One can see that the pore size (both diameter and length) decreases with the increasing loading of bridging components. Overall, the pore diameter for these OSG films follows this sequence: M > E > B for various bridge loadings, except at 45 mol% where 45E > 45M > 45B.

To sum up, the pore size of MSSQ (no carbon bridge) at fixed 30 wt% porogen is the largest because MTMS has a terminal CH_3_ group forming a relatively loose network structure with a large intrinsic volume. Unlike MSSQ with a terminal –CH_3_ group, the organic silica components possess three Si–O bonding and a carbon bridge to Si, i.e., more network linkages, resulting in smaller intrinsic volume, thus pore size, with increasing loading of carbon bridges from 25 to 100 mol%. The flexibility of carbon bridge molecules and their loading relative to MTMS also affects their pore size, and possibly ordering. Similar to silicon oxide tetrahedra, carbon-bridged siloxanes tetrahedra all have three identical corner configurations (Si–O–Si) and their fourth tetrahedral corner are interconnected by:–CH_2_–segment with two rotational axes for BTESM;–CH_2_–CH_2_– segment with three rotational axes for BTMSE;1,4-phenylene unit with Si in a para-configuration having only one rotational axis for BTESB.

Thus, the order, in terms of rotational freedom, of these tetrahedra is B < M < E. In other words, among three bridging components, 1,4-phenylene bridge is the most rigid with little rotational freedom. Combined with its spacing effect, the 1,4-phenylene rings in “B” films tend to align due to π–π stacking, resulting in the smallest pore size. When the loading of carbon-bridged siloxanes is small, say 25 mol%, the empty space in the fixed 75 mol% MTMS and porogen enables the structural adaptation of all three carbon bridges with different rotational freedom into a short-range pore ordering, previously illustrated in Figure 10b–d and Figure 11a–c. When their loading is increased to 45 mol%, only BTMSE with the best flexibility still retains the mobility for structural adaption within MTMS, leading to a short-range pore ordering. No short-range pore ordering is observed for loading ≥60 mol%, because the difference in the rotational freedom does not affect much in a relatively high cross-linked structure.

The data obtained by GISAXS and EP show a relatively good correlation, especially when taking into account that they are based on completely different physical principles. Table 2 shows the pore diameters calculated from GISAXS and EP data. They show the same tendency and are even qualitatively similar. The most pronounced differences show 1,4-phenylene-bridged Sample 45B and Sample 60B (Table 2). According to EP, these samples are perfectly microporous. One can assume that these films have some additional internal gradients able to differently scatter X-rays. GISAXS makes it possible to measure pore length, which well correlates with pore size: the pores with small pore diameter are shorter. The pore-to-pore distance was calculated for the samples with ordered pores. One can see that that this distance is proportional to the pore diameter. It is also phenomenologically understandable. If one is to imagine two materials with cylindrical pores, the number of pores will be larger when the pore diameter is smaller. Therefore pore-to-pore distance will also follow this tendency.

Generally, both EP and GISAXS show that pore size and porosity decrease from methyl-terminated OSG to methylene/ethylene and 1,4-phenylene-bridged films. The pore diameter for these OSG films follows this sequence: MSSQ > M ≈ E > B. A separate study of the films shrinkage happening during the final curing (Table 1) showed that the changes of thicknesses ∆*d* were equal to 15% (MTMS), 26% (100M), 27% (100E) and 32% (100B). It may suggest that the change of thickness and the porosity reduction is mainly happening during the final curing.

### 3.5. Bridge Destruction by Residual Oxygen (DFT Data)

Thermal stability of low-*k* components is an important issue for their integration. For instance, low-*k* films need to be cured at a temperature of about 400–450 °C and the further integration procedure also involves thermal annealing steps (for instance, diffusion barrier deposition). Relatively low thermal stability of carbon-bridged PMO materials is known [42]. Moreover, we recently found that 1,4-phenylene rings can be more sensitive to thermal destruction in the presence of UV light even with λ > 200 nm [24]. It was shown that this effect is related to a shift of absorption edge in the UV region of 1,4-phenylene-bridged OSG to higher λ in comparison with ethylene/methylene-bridged [30] and methyl-terminated materials. It has also been shown that all these bridges are sensitive to the curing environment. The presence of oxygen traces during the curing can significantly destroy the carbon bridges [43]. Here we studied the mechanism of destruction of methylene, ethylene and 1,4-phenylene bridges by residual oxygen. We believe that this knowledge would be useful for the further optimization of technological processes.

DFT analysis is done on the model polymers with –Si–CH_2_–Si–, –Si–CH_2_–CH_2_–Si– and –Si–1,4-Ar–Si– bridge groups (shown in Figure 13, Ar is aryl) to analyze the most energetically favorable reaction paths occurring during the oxidation of the studied bridges by oxygen. The results of calculations are given in Figure 14, Figure 15 and Table 3.

The structure and electron distribution of the highest occupied molecular orbital (HOMO) responsible for the reactivity of model polymers are given in Figure 13. As can be seen, the HOMO electrons of Structure I and Structure II are localized mostly on the –Si–CH_2_–Si– and –Si–CH_2_–CH_2_–Si– bridge groups, indicating that they are the most reactive places. In the case of 1,4-Ar-bridged film (Structure III), most electron density is localized on C_1_ and C_4_ from the aromatic ring. The positions next to C_1_ and C_4_ on ring are more favorable for reaction with OH radical than C_1_ and C_4_.

The possible reaction pathways that could lead to the decomposition of polymers with different bridge groups (shown in Figure 14 and Figure 15) are multistep reactions. The first step is the abstraction of the hydrogen atom from methylene or ethylene induced by OH radical resulting in the formation of a reactive radical place either on the methylene or ethylene bridges. As can be seen from the Table 3, this reaction is most favorable in the case of the Si–CH_2_–CH_2_–Si– bridge group (Structure II). In the case of the –Si–1,4-Ar–Si– bridge group (Structure III), the abstraction of hydrogen by OH radical is energetically possible from C_2_ on the aromatic ring. This reaction is, however, essentially less favorable in comparison with the methylene or ethylene bridges. The second step is the addition reaction of the oxygen molecule in a ground triplet state on the radical place with the formation of the peroxyl radical. This reaction is energetically favourable in the case of all three studied molecules. In the case of Structure I and Structure II, the next step (Reaction 3, Figure 14) is the exergonic intramolecular H-shift to oxygen with the formation of the terminal easy-bonded OH group. This OH group could be shifted either to Si_1_ or Si_2_ atoms (Reactions 4 and 5, Figure 14). These reactions are strongly exergonic and may result in corresponding bond scission. If Structure I and Structure II are compared, it can be seen that polymers with an ethylene bridge should be less stable than those with a methylene one. For the methylene bridge, a possible alternative pathway of Si–C decomposition is presented as Reaction 6. This reaction is similar to the self-hydrophobization process that was already discussed in Ref. [44,45]. The Reaction 6 is exergonic but might be kinetically suspectable because of the formation of reaction intermediates containing 5-valence C and Si atoms. However, the formation of 5-valence Si atoms as intermediate states was already reported for low-*k* etching by atomic fluorine [46]. Therefore, this mechanism can be considered as a possible way of degradation.

In the case of Structure III, the Reaction 3 may be a H abstraction from the polymer chain by the peroxyl radical. This reaction is, however, energetically unfavourable and could not result further in any bond scission. Otherwise, in the case of Structure III, there may be an alternative pathway through the direct addition of the oxygen molecule in ground triplet state on the aromatic ring followed by an intramolecular H shift to oxygen (Reaction 5, Figure 15). This concerted Reaction 5 is actually a well-known insertion reaction. The formed terminal OH group is easily bonded and could be shifter either to aromatic ring position C_1_ (Reaction 6), or to the Si atom (Reaction 7). Reaction 6 is essentially energetically more favourable than the strong endergonic Reaction 7 and could result in a corresponding bond scission.

From the comparison of energetics of possible reaction pathways for Structures I, II and III, it may be expected that Structure II should be less stable, and Structure III the most stable for the possible polymer decomposition.

## 4. Conclusions

Organosilicate (OSG)-based low dielectic constant films with different ratios of terminal methyl and bridging organic (methylene, ethylene and 1,4-phenylene) groups are spin-on deposited by using mixtures of alkylenesiloxane with different organic bridges (BTESM, BTMSE, and BTESB) and methyltrimethoxysilane (MTMS). The deposited films were first soft baked at 120, 200 °C and then cured at 430 °C. The films’ porosity was controlled by using sacrificial template Brij^®^ L4 (30 wt%). Changes of the films refractive indices (RI), mechanical properties, *k*-values, porosity and pore structure versus chemical composition of the film’s matrix are evaluated. 

Porosity and pore structure are evaluated by using ellipsometric porosimetry (EP) and grazing-incidence small-angle X-ray scattering (GISAXS). According to EP results, the samples prepared from pure MTMS (only terminal methyl groups, no bridging groups) have the highest porosity and pore size, then they gradually decrease with the concentration of the bridging groups’ precursors. The films prepared from only BTESB (the highest concentration of 1,4-phenylene as a bridging group) only have micropores. MTMS additive to BTESB increases the pore size, making them bi-modal. According to GISAXS, the films prepared from 25 mol% of carbon bridged precursors (25M, 25E, and 25B), the pore arrangement has a short-range in-plane ordered state. The films prepared from pure MTMS or higher concentrations of alkylenesiloxane have disordered structures (except Sample 45E). The pore size calcucated from EP and GISAXS data have a reasonably good agreement. Pore-to-pore distance calculated for the films with ordered porosiy can also be qualitatively described by EP measured pore size. The mechanical and dielectric properties of porous hybrid OSG films are determined by their porosity, degree of networking, and carbon bridge components. With an increasing carbon-bridged components loading, an increased degree of networking leads to a higher Young’s modulus, and also an unfavorably higher dielectric constant. The network connectivity is also found to be the primary factor controlling the porosity and pore radius distribution. It is also shown that the bridging carbon groups are not able to make these materials completely hydrophobic and, therefore, the concentration of terminal methyl groups must exceed a certain critical level if the films are targeted for microelectronics application.

The chemical resistance of the films to annealing in an oxygen-containing environment is evaluated by using density functional theory (DFT). Mechanisms of destruction are discussed and it is shown that 1,4-phenylene-bridged films have the highest stability while methylene-bridged films are more stable than ethylene-bridged films. The limited thermal stability of these films in the oxygen-containing environment must be taken into account during the development of new technological processes for their integration into ULSI devices.

Finally, the application of mixtures of alkylenesiloxane with different organic bridges between the silicon atoms (BTESM, BTMSE, and BTESB) with methyl-modified silicon alkoxide (MTMS) makes it possible to deposit OSG low-*k* films with various properties. Their porosity, pore size and ordering, hydrophilicity, mechanical properties and *k*-values can be controlled by selected ratios of alkylenesiloxane and methyl-modified silicon alkoxide. These possibilities make them attractive for microelectronics technology. However, improvement of mechanical properties is often counteracted by the higher *k*-value of the films containing organic bridges. This observation can be explained by an increase in the matrix density when Si–O–Si is replaced with the Si–CH_2_–Si groups. Therefore, the optimal precursors ratio must be carefully selected for each technological application.

## Figures and Tables

**Figure 1 materials-13-04484-f001:**
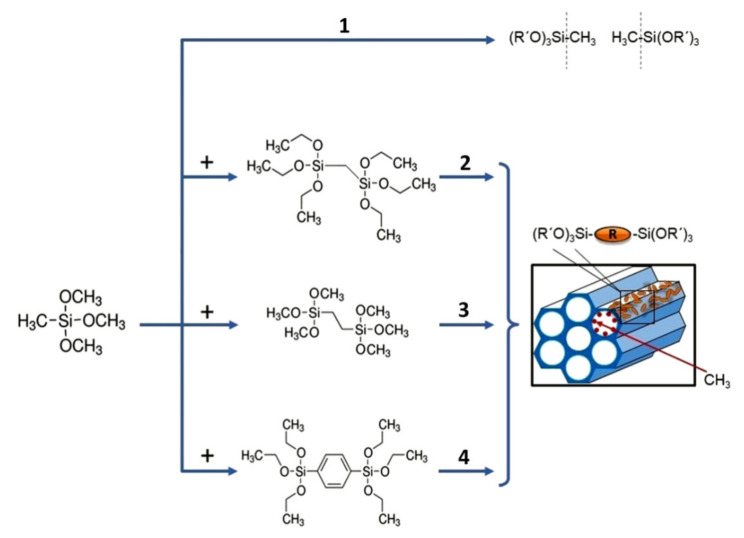
Porous organosilicate glass films prepared by using only methyltrimethoxysilane (**1**) or its mixture with one of the following alkylenesiloxane: 1,2-bis(triethoxysilyl)methane (**2**), 1,2-bis(trimethoxysilyl)ethane (**3**) or 1,4-bis(triethoxysilyl)benzene (**4**). After evaporation-induced self-assembly (EISA) deposition, the bridging carbon groups are located on the pore wall while the terminal methyl groups cover the pore’s internal surface.

**Figure 2 materials-13-04484-f002:**
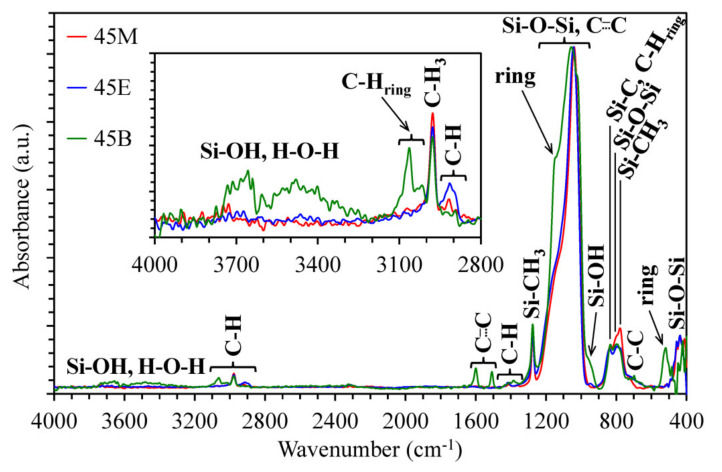
FTIR spectra of periodic mesoporous organosilica films with different bridges (45M—methylene, 45E—ethylene, 45B—1,4-phenylene), 45 mol%, annealed at 430 °C for 30 min in air (hard bake).

**Figure 3 materials-13-04484-f003:**
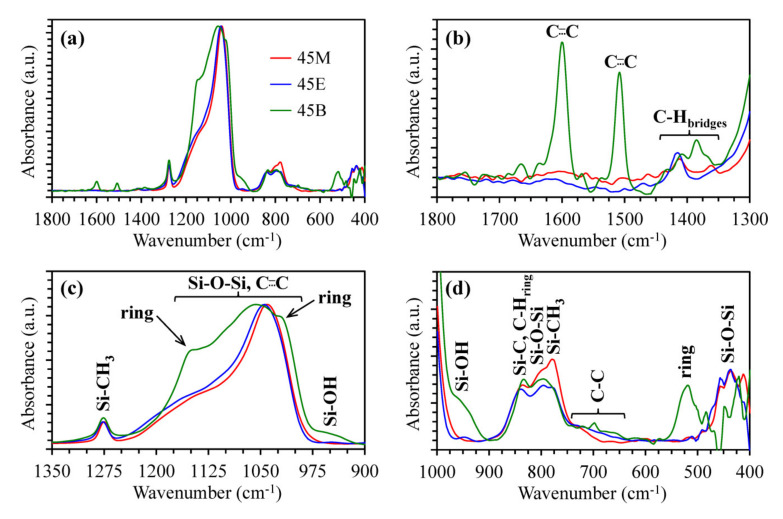
Entire fingerprint region (**a**) and its local ranges (**b**–**d**) of FTIR spectra of hard baked (at 430 °C in air) periodic mesoporous organosilicate films with different bridges.

**Figure 4 materials-13-04484-f004:**
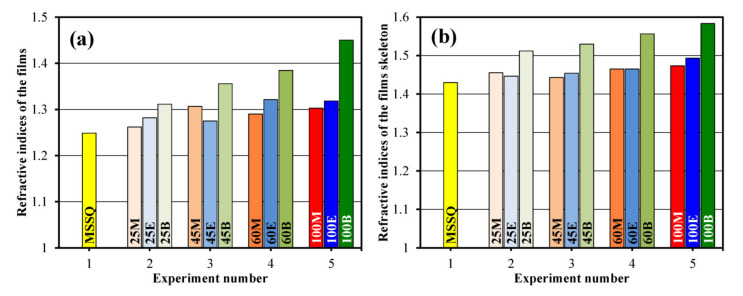
Refractive indices of the studied films (**a**) and their skeleton (**b**). The column termed as Experiment 1 corresponds to the methylsilsesquioxane (MSSQ) film prepared from pure methyltrimethoxysilane (MTMS). The samples in Experiments 2, 3, 4 correspond to concentrations of bridging groups’ precursors equal to 25, 45 and 60 mol%, respectively. No MTMS was used in Experiment 5.

**Figure 5 materials-13-04484-f005:**
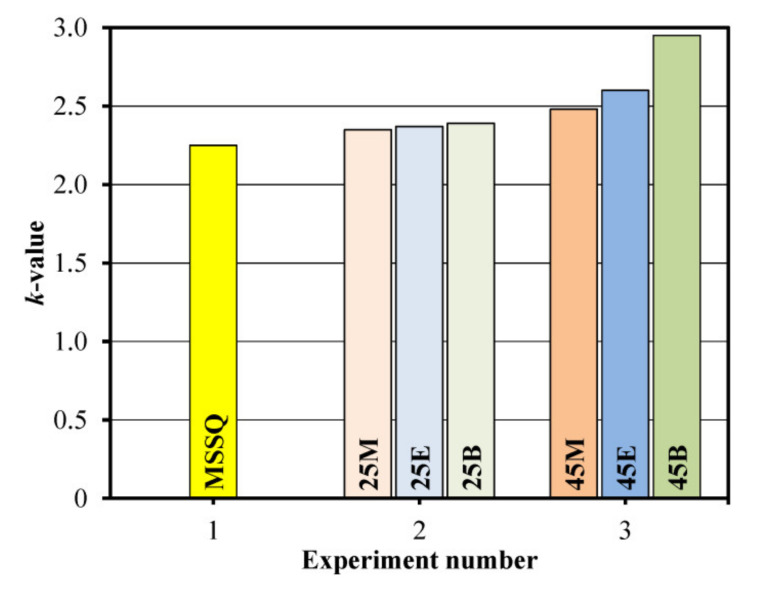
Dependence of the measured *k*-value (at 100 kHz) on the film composition. The column termed as Experiment 1 corresponds to the methylsilsesquioxane (MSSQ) film prepared from pure methyltrimethoxysilane (MTMS). The samples in Experiments 2, 3 correspond to concentrations of bridging groups’ precursors equal to 25 and 45 mol%, respectively.

**Figure 6 materials-13-04484-f006:**
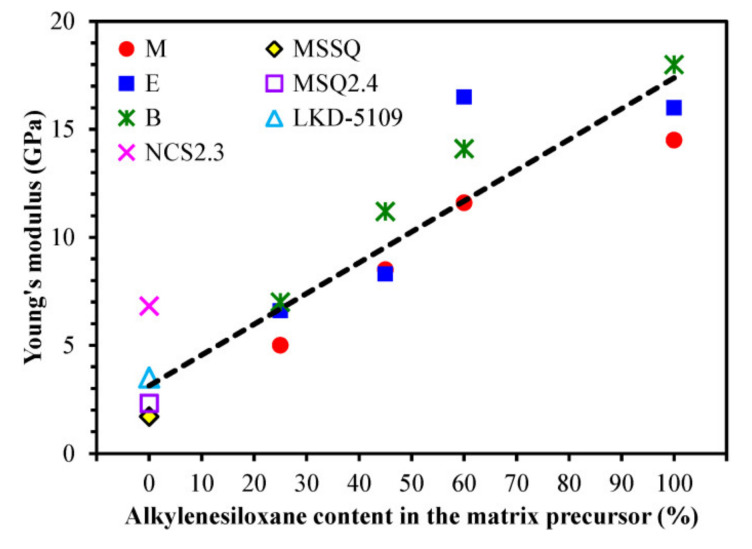
Change of Young’s modulus (YM) versus alkylenesiloxane content in the matrix precursor. The YM of porous organosilicate glass low-*k* films with only terminal carbon groups are represented by the methylsilsesquioxane (MSSQ) sample (*V* = 39%). The YM values reported for well-known low-*k* films with similar porosity: MSQ2.4 (*V* = 42%) [37], LKD-5109 (*V* = 39%) [38] and NCS2.3 (*V* = 33%) [37] are included for comparison. In this Figure: M—methylene bridge (BTESM), E—ethylene bridge (BTMSE), B—1,4-phenylene bridge (BTESB).

**Figure 7 materials-13-04484-f007:**
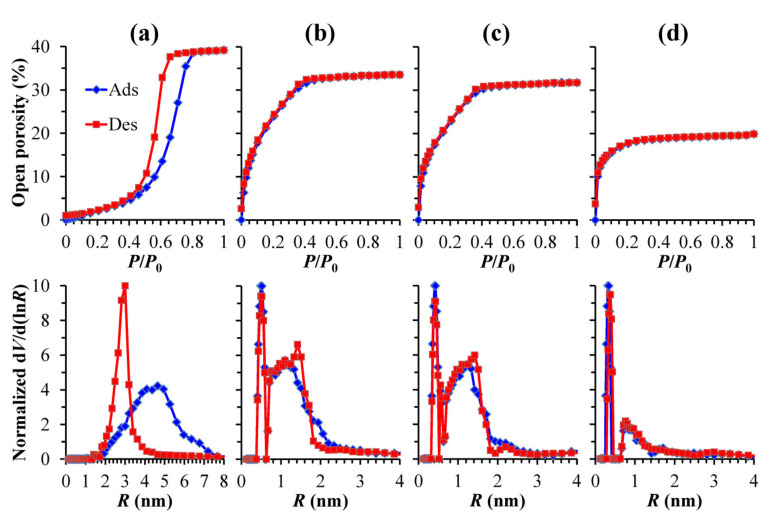
Adsorption–desorption isotherms and pore size distribution in the low-*k* films prepared from pure: (**a**) methyltrimethoxysilane (MTMS), (**b**) 1,2-bis(triethoxysilyl)methane (BTESM), (**c**) 1,2-bis(trimethoxysilyl)ethane (BTMSE) and (**d**) 1,4-bis(triethoxysilyl)benzene (BTESB) at a fixed 30 wt% porogen content.

**Figure 8 materials-13-04484-f008:**
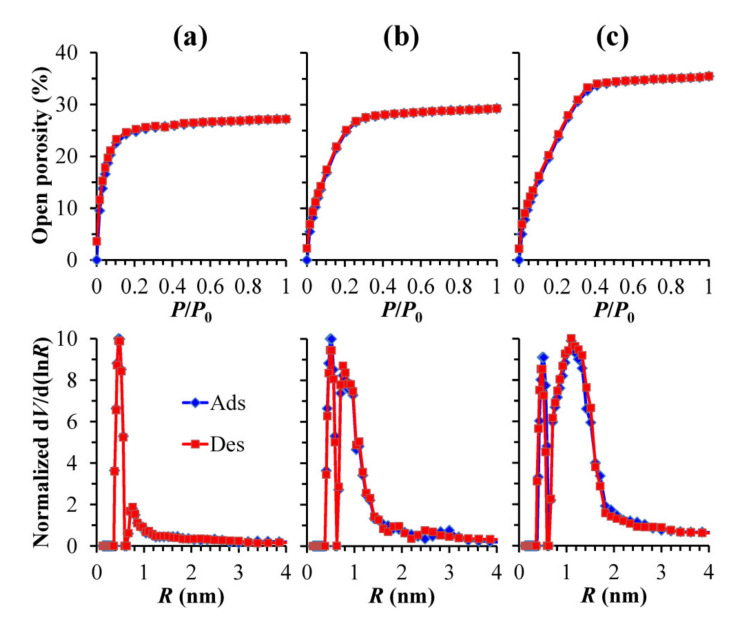
Adsorption–desorption isotherms of isopropyl alcohol vapor and pore size distribution in low-*k* films deposited with different ratios of methyltrimethoxysilane to 1,4-bis(triethoxysilyl)benzene (MTMS/BTESB) mixture: (**a**) 40/60, (**b**) 55/45 and (**c**) 75/25.

**Figure 9 materials-13-04484-f009:**
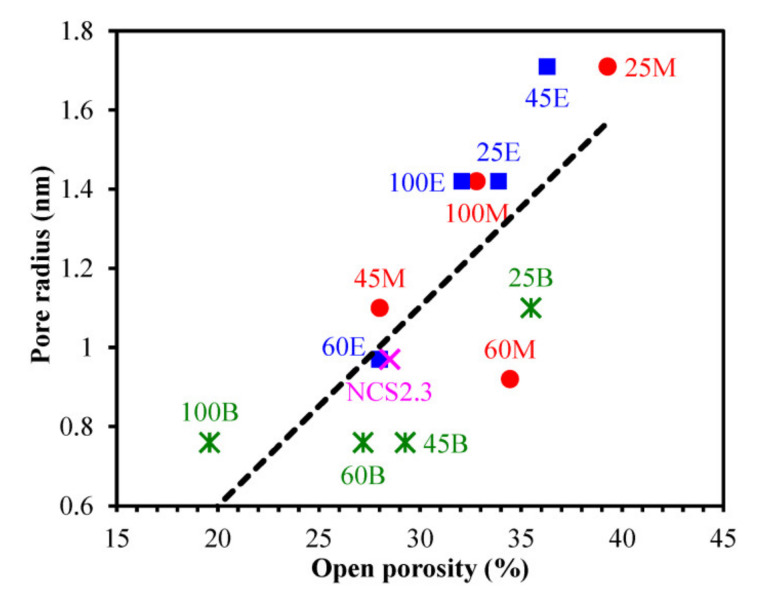
Pore radius of studied organosilicate films versus open porosity (ellipsometric porosimetry data). In this Figure: M—methylene bridge (BTESM), E—ethylene bridge (BTMSE), B—1,4-phenylene bridge (BTESB) and preceding digits indicate the alkylenesiloxane content in the matrix precursor in mol%. The data point of NCS2.3 low-*k* film is included for comparison.

**Figure 10 materials-13-04484-f010:**
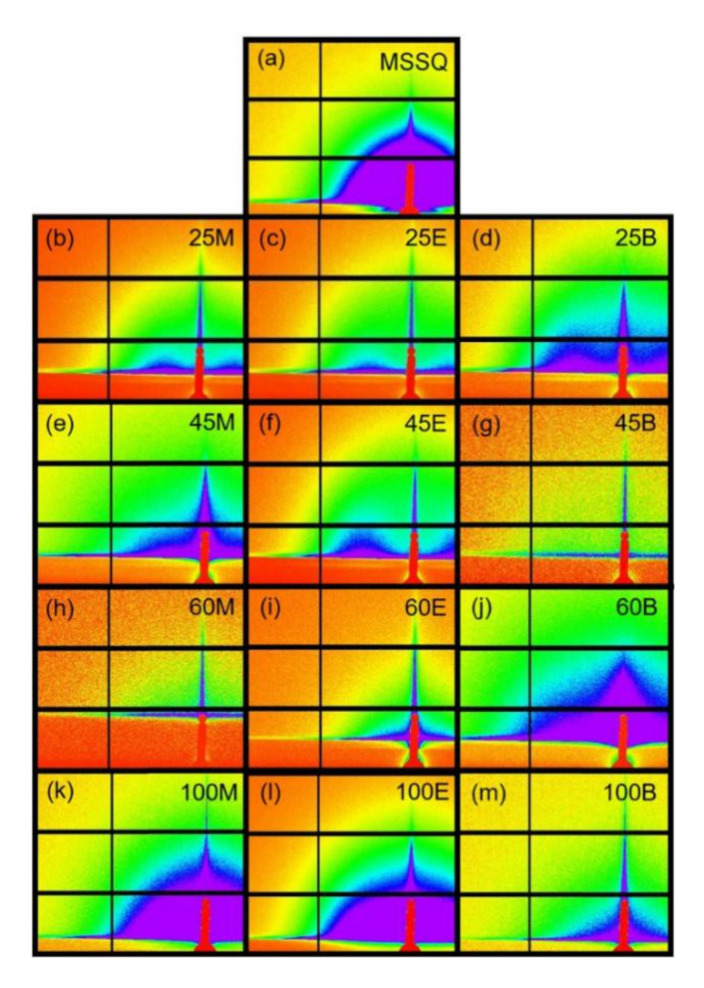
Grazing-incidence small-angle X-ray scattering patterns of porous organosilicate glass films: (**a**) methylsilsesquioxane (MSSQ); (**b**) 25M; (**c**) 25E; (**d**) 25B; (**e**) 45M; (**f**) 45E; (**g**) 45B; (**h**) 60M; (**i**) 60E; (**j**) 60B; (**k**) 100M; (**l**) 100E; (**m**) 100B, where M is the methylene bridge (BTESM), E—ethylene bridge (BTMSE), B—1,4-phenylene bridge (BTESB).

**Figure 11 materials-13-04484-f011:**
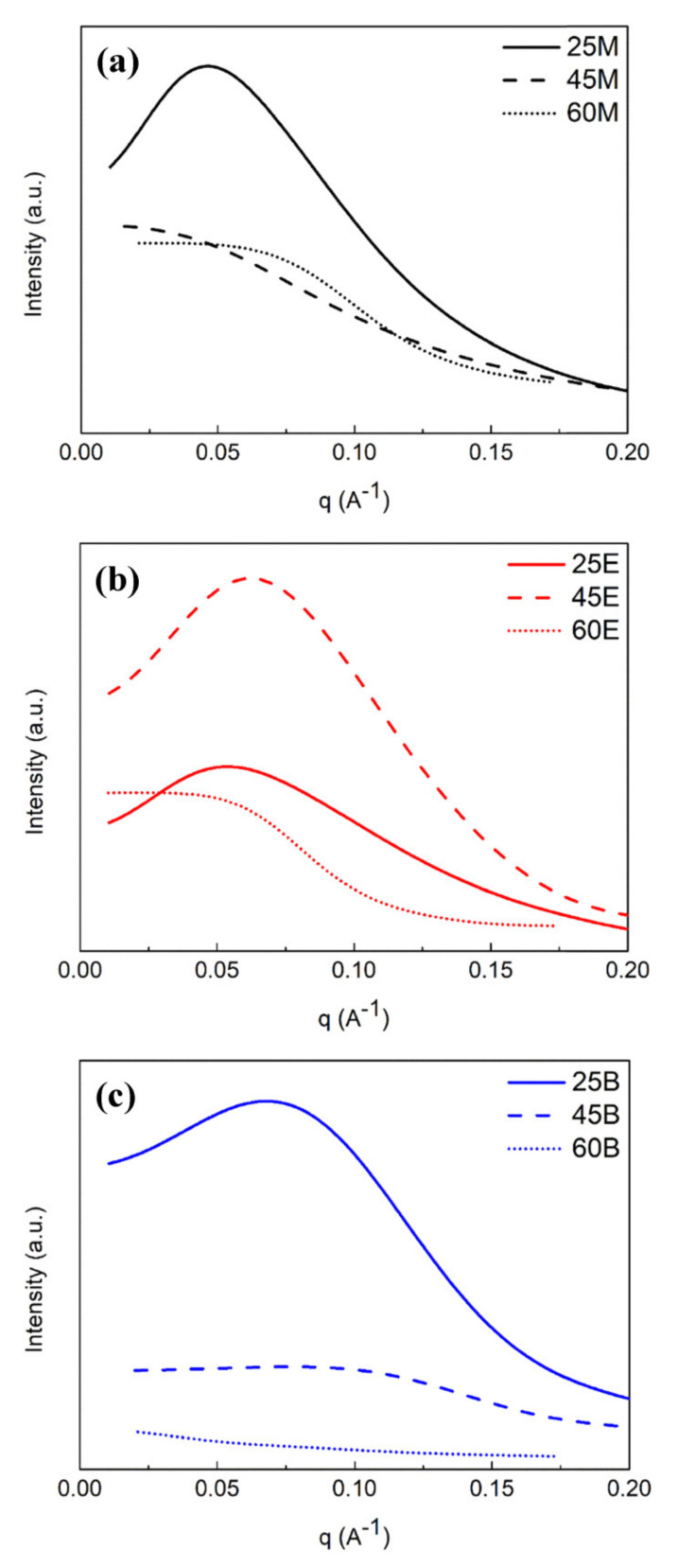
Pore correlation pattern GISAXS in-plane intensity versus *q_xy_* for: (**a**) 25/45/60M; (**b**) 25/45/60E; (**c**) 25/45/60B hybrid organosilicate glass films.

**Figure 12 materials-13-04484-f012:**
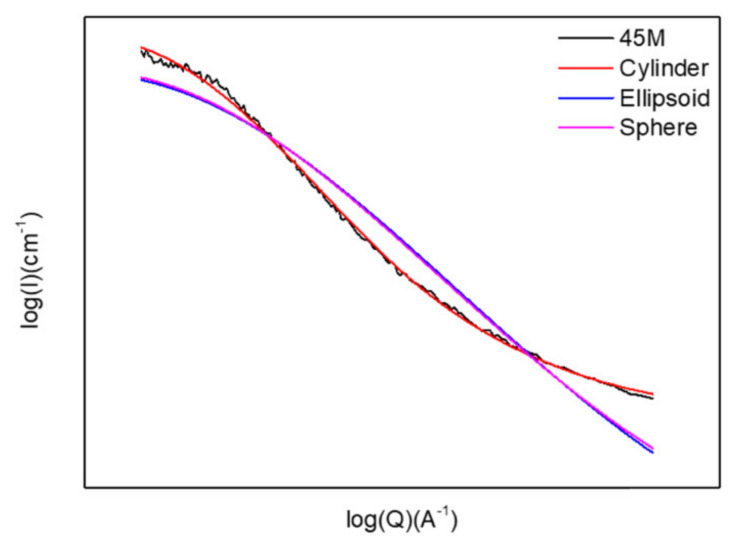
Intensity versus *q_xy_* for the 45M organosilicate glass film and the fitting curves of the cylinder, ellipsoid and sphere pore models, where M is the methylene bridge (BTESM).

**Figure 13 materials-13-04484-f013:**
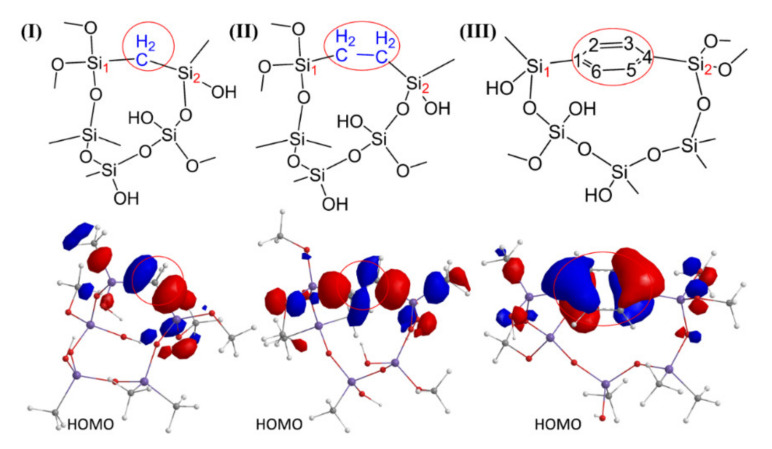
Structures of model polymers with (**I**) methylene-, (**II**) ethylene- and (**III**) 1,4-Ar bridges (Ar is aryl) used for calculations, and electron distribution of the highest occupied molecular orbital (HOMO) responsible for the reactivity of the model polymers.

**Figure 14 materials-13-04484-f014:**
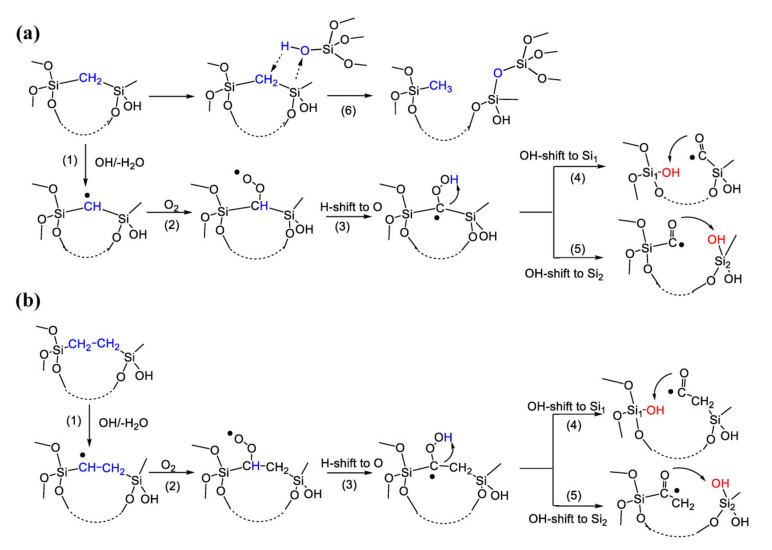
Calculated energetically favourable possible reaction pathways leading to Si–C bond scission in model polymers with –Si–CH_2_–Si– (**a**) and –Si–CH_2_–CH_2_–Si– (**b**) bridges. Structures of model polymers are simplified for clarity. Reaction parameters Δ*H* and Δ*G* are given in Table 3.

**Figure 15 materials-13-04484-f015:**
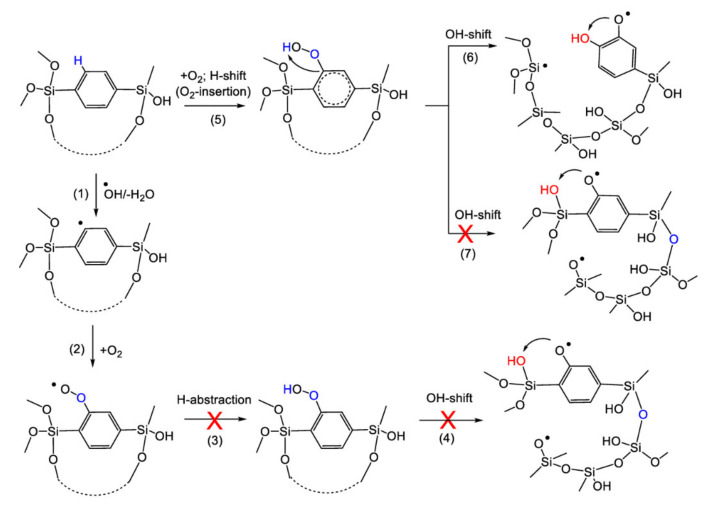
Calculated energetically favourable possible reaction pathways which could lead to bond scission in model polymers with the –Si–1,4-Ar–Si– bridge. Structures of model polymers are simplified for clarity. Reaction parameters Δ*H* and Δ*G* are given in Table 3.

**Table 1 materials-13-04484-t001:** Characteristics of porous methylsilsesquioxane film (MSSQ) and organosilicate glass films with different types (M—methylene, E—ethylene, B—1,4-phenylene) and content (0–100 mol%) of bridging groups, annealed at 430 °C for 30 min in air (hard bake). In this Table: MTMS is methyltrimethoxysilane, BTESM is 1,2-bis(triethoxysilyl)methane, BTMSE is 1,2-bis(trimethoxysilyl)ethane, BTESB is 1,4-bis(triethoxysilyl)benzene.

Exp. No	Sample No	Sol Composition (mol%)	Thickness *d* (nm)	Shrinkage Δ*d* (%)	Refractive Index	Pore Radius (nm)	Porosity (%)	Young’s Modulus (GPa)	*k*-Value	Skeleton Refractive Index
1	MSSQ	MTMS = 100	298	15	1.25	3.00	39.2	1.7 *	2.25	1.43
2	25M	MTMS = 75 BTESM = 25	243	14	1.26	1.71	39.3	5.0	2.35	1.46
	25E	MTMS = 75 BTMSE = 25	238	20	1.28	1.42	33.9	6.6	2.37	1.45
	25B	MTMS = 75 BTESB = 25	312	19	1.31	1.10	35.5	7.0	2.39	1.51
3	45M	MTMS = 55 BTESM = 45	260	16	1.31	1.10	28.0	8.5	2.48	1.44
	45E	MTMS = 55 BTMSE = 45	419	19	1.28	1.71	36.3	8.3	2.60	1.45
	45B	MTMS = 55 BTESB = 45	160	17	1.36	0.76	29.3	11.2	2.95	1.53
4	60M	MTMS = 40 BTESM = 60	327	27	1.29	0.92	34.4	11.6	*-*	1.47
	60E	MTMS = 40 BTMSE = 60	227	30	1.32	0.97	28.0	16.5	*-*	1.47
	60B	MTMS = 40 BTESB = 60	110	38	1.39	0.76	27.2	14.1	*-*	1.56
5	100M	MTMS = 0 BTESM = 100	183	26	1.30	1.42	32.8	14.5	*-*	1.47
	100E	MTMS = 0 BTMSE = 100	203	27	1.32	1.42	32.1	16.0	*-*	1.49
	100B	MTMS = 0 BTESB = 100	162	32	1.45	0.76	19.6	18.0	*-*	1.58

* the Young’s modulus (YM) value of MSSQ sample was measured by AFM [34].

**Table 2 materials-13-04484-t002:** Pore geometry (cylinder disc pore diameter, length, aspect ratio) and pore-to-pore distance of porous methylsilsesquioxane film (MSSQ) and organosilicate glass films with different types (M—methylene, E—ethylene, B—1,4-phenylene) and content (0–100 mol%) of bridging groups. Ellipsometric porosimetry (EP) pore diameters are included for comparison. In this Table: MTMS is methyltrimethoxysilane, BTESM is 1,2-bis(triethoxysilyl)methane, BTMSE is 1,2-bis(trimethoxysilyl)ethane, BTESB is 1,4-bis(triethoxysilyl)benzene.

Exp. No	Sample No	Sol Composition (mol%)	Grazing-incidence Small-angle X-ray Scattering Data				EP Pore Diam. (nm)
			Diam. (nm)	Length *L* (nm)	Aspect Ratio (diam./length)	Pore-to-pore Distance (nm)	
1	MSSQ	MTMS = 100	4.6	3.4	1.35	-	6.00
2	25M	MTMS = 75 BTESM = 25	3.8	2.2	1.73	19.0	3.42
	25E	MTMS = 75 BTMSE = 25	2.6	1.2	2.17	10.8	2.84
	25B	MTMS = 75 BTESB = 25	2.2	1.0	2.20	8.1	2.20
3	45M	MTMS = 55 BTESM = 45	2.0	0.6	3.33	-	2.20
	45E	MTMS = 55 BTMSE = 45	3.4	2.0	1.70	8.6	3.42
	45B	MTMS = 55 BTESB = 45	3.2	1.8	1.78	-	1.52
4	60M	MTMS = 40 BTESM = 60	2.0	1.0	2.00	-	1.84
	60E	MTMS = 40 BTMSE = 60	1.8	0.6	3.00	-	1.94
	60B	MTMS = 40 BTESB = 60	3.0	1.5	2.00	-	1.52
5	100M	MTMS = 0 BTESM = 100	2.0	0.8	2.50	-	2.84
	100E	MTMS = 0 BTMSE = 100	1.8	0.8	2.25	-	2.84
	100B	MTMS = 0 BTESB = 100	1.6	0.5	3.20	-	1.52

**Table 3 materials-13-04484-t003:** Reaction parameters Δ*H* and Δ*G* in kcal∙mol^−1^ according to the reaction pathways given in the reaction schemes in Figure 14 and Figure 15.

Reaction	–Si–CH_2_–Si–	–Si–CH_2_–CH_2_–Si–	–Si–1,4-Ar–Si–
1	Δ*H* = −13 Δ*G* = −15	Δ*H* = −26 Δ*G* = −27	Δ*H* = −3 Δ*G* = −4
2	Δ*H* = −19 Δ*G* = −11	Δ*H* = −26 Δ*G* = −14	Δ*H* = −47 Δ*G* = −34
3	Δ*H* = −13 Δ*G* = −15	Δ*H* = −32 Δ*G* = −32	Δ*H* = +15 Δ*G* = +15
4	Δ*H* = −63 Δ*G* = −63	Δ*H* = −51 Δ*G* = −50	Δ*H* = +18 Δ*G* = +17
5	Δ*H* = −57 Δ*G* = −57	Δ*H* = −46 Δ*G* = −46	Δ*H* = −26 Δ*G* = −13
6	Δ*H* = −18 Δ*G* = −20	-	Δ*H* = −5 Δ*G* = −8
7	-	-	Δ*H* = +25 Δ*G* = +22

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
