# Peer review of "Effects of Methyl Terminal and Carbon Bridging Groups Ratio on Critical Properties of Porous Organosilicate Glass Films"

_materials, 2020, doi:10.3390/ma13204484_

Round 1
Reviewer 1 Report
The manuscript details the synthesis and characterization of microporous/mesoporous organosilica films. The results are well discussed and they could merit publication, provided the following points are addressed:
The novelty of the work, especially in relation with previous studies must be highlighted. There are many examples in the literature of PMOs with the same organic groups as those presented herein. Please revise the introduction section to include relevant literature examples of methylene, ethylene, 1,4-phenylene PMOs and the novelty of this manuscript.
The authors state in the introduction that hydrophobic character is important for low k films. This is not experimentally determined. The authors are instructed to carry out thermogravimetric analyses up to 800 – 1000 °C and to quantify both the amount of physisorbed water and to experimentally determine the thermal stability of the films. The amount of physisorbed water can be correlated with the ~1640 cm-1 IR vibration. The hydrophobic/hydrophilic character of the films can be determined through water contact angle experiments.
In addition the correlation of experimentally determined thermal stability with the ΔH / ΔG values obtained from DFT could be performed.
P.2. “The final Si content was mSi/msol” The msol abbreviation must be defined in text.
p.5. The adsorbed water in 45B can also be noticed from the ~1640 (1645) cm-1 IR vibration. The 3200 -3600 could correspond also to silanol O-H vibrations, not physisorbed water so they are not as conclusive. Please revise accordingly.
Author Response
Dear Reviewer,
Thanks for your comments and questions.
Our response to your comments you can find in the attached file.
Please see the attachment.

Reviewer 2 Report
The present manuscript by Vishnevskiy et al. deals with porous organosilane films which are prepared by condensation of trialkoxysilanes with different bridging groups. The paper is well written, and clearly addresses relevant issues in the field of low-K dielectrics.
There are several points that must be clarified before a publication can be considered.
page 2: the sentence "Therefore, simultaneous introduction of terminal groups and bridging carbon groups is also needed is misleading - omit.
page 2: the sample preparation remains unclear. If a recipe is provided in the literature (e.g., in another article of the authors) that must be specified clearly, so that the reader can follow the sample preparation procedure.
What are "special precursors"? It´standard chemicals from commercial sources!
The compound MTMS is termed "condensation methyl-modified silicon alkoxide" - this term is not appropriate
page 3: in the formula graphics the term MSSQ appears, but no reference is made to it in the text. I assume this would be MTMS.
I have some concerns about the Interpretation of FTIR spectra:
page 4: in the text related to FTIR spectra the authors state that "the peaks related to the bridgig Groups are not so pronounced because of their non-polar structure. What do the authors want to express ? Hydrocarbons give quite srong IR sigals although they are non-polar compounds.
Figure 2 and Figure 3: here, the FTIR signals at 1500 cm-1 and 1600 cm-1 are attributed to C=C double bonds (which are not present in the structures. These signals are attributable to aromatic ring vibrations!
1,4-disubstituted aromatic compounds Show a strong signal in the spectral range 800 - 840 cm-1. Why do the authors attribute the signal at 500 cm-1 to structure B containing the 1-4 linked aromatic unit? Is there a reference spectrum, or a literature reference for this?
The authors conclude that sample 45 B contains more water than the other structures. I don´t think that this can be concluded from the FTIR spectra.
page 16:
The UV sensitivity of aromatic components should be discussed properly. As benzene rings absorb UV light up to 290 nm, it is no surprise that they will be sensitive to (prolonged) short wavelength Irradiation. This is well known from, e.g., polystyrene and polycarbonate.
page 17, Figure 14 and Figure 15: regarding the proposed degradation pathways I am missing literature refrences which would substantiate the individual reactions. In the present manuscript it remains unclear which mechanisms are proven, and which are speculative.
Conclusion (and abstract): the finding that aromatic structures are more stable against oxidative degradation is not surprising. This is well known from the chemistry of photoresists, and the interaction of O2 plasmas with aromatic and aliphatic polymers in electronics industry. Please comment on this.
Author Response
Dear Reviewer,
Our explanations and answers on your comments and questions are given below. All changes in the main text were marked.
Our response to your comments you can find in the attached file.
Please see the attachment.

Reviewer 3 Report
The manuscript presents successful fabrication of porous low dielectic organosilicate films. The authors demonstrated that simple and environmentally friendly process was employed for the fabrication. The authors 욘systemically demonstrated that dielectric indices, mechanical properties, k-values, porosity and pore structure was controlled by changing bridging organic (methylene, ethylene and 1,4-phenylene) groups. These results were fully helpful to the researcher in the application field. Therefore, it is suitable for publication in materials. However, minor corrections need prior to publication: (1) For real application, uniformity of obtained films also important factor. So, authors should insert the SEM and UV-vis transmittance spectra of obtained films. (2) When bridging organic (methylene, ethylene and 1,4-phenylene) groups were changed, porosity and pore structure were changed. Is there any relationship between porosity and pore structure and dielectric constant? If yes, should be inserted into the discussion parts. (3) What is hydrophilicity of each obtained films?
Author Response
Dear Reviewer,
Our explanations and answers on your questions are given below. All changes in the main text were marked.
Our response to your comments you can find in the attached file.
Please see the attachment.

Round 2
Reviewer 1 Report
The authors have performed most required modifications, answered all concerns and the quality of their manuscript has improved. The manuscript can be accepted for publication in its current form.
Regarding the TGA analysis, the 800 - 1000 °C maximum temperature is necessary for water/silanol quantification since silica materials gradually lose water up to those temperatures. The authors are correct in pointing out that this technique is destructive to the material. It is unfortunate that the authors could not perform these measurements, however such situation is understandable. It is also true that mass losses lower than around 1% cannot be accurately determined by thermogravimetric analyses.